
# *Escherichia coli* concentration, multiscale monitoring over the decade 2011-2021 in the Mekong basin, Lao PDR

Laurie BOITHIAS[1], Olivier RIBOLZI[1], Emma ROCHELLE-NEWALL[2], Chanthanousone THAMMAHACKSA[3], Paty NAKHLE[1], Bounsamay SOULILEUTH[3], Anne PANDO-BAHUON[3], Keooudone LATSACHACK[3], Norbert SILVERA[3], Phabvilay SOUNYAFONG[3], Khampaseuth XAYYATHIP[3], Rosalie ZIMMERMANN[4,5,6], Sayaphet RATTANAVONG[4], Priscia OLIVA[1], Thomas POMMIER[7], Olivier EVRARD[8], Sylvain HUON[2], Jean CAUSSE[9], Thierry HENRY-DES-TUREAUX[3], Oloth SENGTAHEUANGHOUNG[10], Nivong SIPASEUTH[10], Alain PIERRET[3]

[1]Géosciences Environnement Toulouse (GET), Université de Toulouse, CNRS, IRD, UPS, Toulouse, France
[2]Sorbonne Université, Univ. Paris Est Creteil, IRD, CNRS, INRAE, Institute of Ecology and Environmental Sciences of Paris (iEES-Paris), Paris, France
[3]IRD, Department of Agricultural Land Management (DALaM), Vientiane, Lao PDR
[4]Lao-Oxford-Mahosot Hospital-Wellcome Trust Research Unit, Microbiology Laboratory, Mahosot Hospital, Vientiane, Lao PDR
[5]Department of Environmental Sciences, University of Basel, Basel, Switzerland
[6]Department of Medical Microbiology and Infection Prevention, Amsterdam University Medical Centers (UMC), Amsterdam, The Netherlands
[7]Laboratoire d'Ecologie Microbienne (LEM), CNRS, UCBL, VetAgroSup, Université de Lyon, Villeurbanne, France
[8]Laboratoire des Sciences du Climat et de l'Environnement (LSCE/IPSL), CEA, CNRS, UVSQ, Université Paris-Saclay, Gif-sur-Yvette, France
[9]Société Transcender, Rennes, France / Ecole des Hautes Etudes en Santé Publique (EHESP), Laboratoire d'Etude et de Recherche en Environnement et Santé, IRSET-INSERM, Rennes, France
[10]Ministry of Agriculture and Forestry (MAF), Department of Agricultural Land Management (DALaM), Vientiane, Lao PDR

*Correspondence to*: Laurie BOITHIAS (laurie.boithias@get.omp.eu) and Olivier RIBOLZI (olivier.ribolzi@ird.fr)

**Abstract.** Bacterial pathogens in surface waters may threaten human health, especially in developing countries, where untreated surface water is often used for domestic needs. The objective of the long-term multiscale monitoring of *Escherichia coli* concentration in stream water, and that of associated variables (temperature, electrical conductance, dissolved oxygen concentration and saturation, pH, oxidation-reduction potential, turbidity, and total suspended sediment concentration), was to identify the drivers of bacterial dissemination across tropical catchments. This data description paper presents three datasets (see section Data availability) collected at 31 sampling stations located within the Mekong river and its tributaries in Lao PDR (0.6-25,946 km²) from 2011 to 2021. The 1,602 records have been used to describe the hydrological processes driving in-stream *Escherichia coli* concentration during flood events, to understand land-use impact on bacterial dissemination on small and large catchment scales, to relate stream water quality and diarrhea outbreaks, and to build numerical models. The database may be further used e.g. to interpret new variables measured in the monitored catchments, or to map the health risk posed by fecal pathogens.





## 1 Introduction

Bacterial pathogens, including fecal bacteria, are etiological agents of several waterborne diseases such as diarrhea. Primary sources of fecal bacteria in the environment are cattle droppings or human feces where open defecation is practiced or where sanitation systems are lacking or deficient (Exley et al., 2015; Tong et al., 2016; Rochelle-Newall et al., 2015). Fecal bacteria

may threaten human health if present in surface water, especially in developing countries where population often uses untreated surface water for domestic needs (Boithias et al., 2016; Vos et al., 2020).

Prior to 2010, to our knowledge, no data had been published to document the occurrence of Fecal Indicator Bacteria (FIB), such as *Escherichia coli* (*E. coli*), in stream water of rural, tropical catchments, such as the area of the 800,000 km$^2$ Mekong river basin. Besides, very little information existed on microbial contamination and dissemination mechanisms in tropical

environments. Accordingly, our research group initiated the systematic monitoring of surface water quality in 2011, focusing on Lao PDR, a landlocked country that contributes 41 and 54% of the total Mekong flow during the dry and the rainy seasons, respectively (MRC, 2009), and where about 67% of the population lives in rural areas (Lao Statistics Bureau, 2015).

The objective of the spatial and temporal monitoring of *E. coli* concentrations in stream water, and that of associated physico-chemical measurements (i.e., temperature, electrical conductance at 25 °C, dissolved oxygen concentration and saturation, pH,

oxidation-reduction potential, turbidity, and total suspended sediment concentration), was to understand bacterial fate and transport, and underlying drivers, during storm and inter-storm flow periods within tropical catchments. The study design included three monitoring scales:

(1) A spatial survey conducted during both the dry and the rainy seasons in the Mekong river (6 sampling stations) and its tributaries (23 sampling stations) across Lao PDR. The survey aimed at assessing the spatial variability of in-

stream *E. coli* concentration at large spatial scales (239-25,946 km² for the tributaries, up to 549,055 km² for the Mekong itself), during both low flow and high flow periods. Grab samples were collected twice: in March and in July 2016;

(2) A temporal monitoring at the outlet of six catchments (6 sampling stations) in the mountainous area of northern Lao PDR, initiated in 2011, to assess the temporal variability of in-stream *E. coli* concentration at large spatial scales (239-

25,946 km² for the tributaries, up to 272,155 km² for the Mekong itself), during both low flow and high flow periods. Grab samples were first collected with a biweekly time-interval, thereafter modified to a 10-day time-interval in 2017;

(3) A temporal monitoring at the outlet of a 0.6 km² headwater catchment (1 sampling station) in the mountainous area of northern Lao PDR, initiated in 2011, which aimed at understanding the dynamics and the drivers of *E. coli* dissemination during low flow and flood events. Grab samples were collected during low flow with a biweekly time-

interval, thereafter modified to a 10-day time-interval in 2017. During flood events, water sample collection was made with an automatic sampler triggered by water level change.





## 2 Methods

### 2.1 Study site

The Mekong river basin has experienced a dramatic economic and population growth over the past half-century, increasing
pressure on land, water, and other natural resources (Pokhrel et al., 2018; Arias et al., 2014; Global Water Forum, 2015). Rapid land-use change, such as deforestation (Lyon et al., 2017) and conversion from traditional slash-and-burn agricultural systems to tree plantations (Ribolzi et al., 2017), have put soil fertility, agricultural productivity, and biodiversity conservation at stake (MA, 2005). Since the 1970s, dams have been constructed on the Mekong river and its tributaries (WLE, 2017), leading to severe impacts on hydrology (Le Meur et al., 2021; Hecht et al., 2019), on sediment transport (Kondolf et al., 2014; Shrestha
et al., 2018), and on aquatic biodiversity (Sabo et al., 2017), in a climate change context.

The 31 sampling stations (Fig. 1, Table 1) of this dataset are located in Lao PDR, within the Mekong river basin. The choice of sampling stations in Mekong tributaries was made to encompass a broad range of catchment sizes (0.6-25,946 km²), and a large range of geological, topographical, and land-use features. In Lao PDR, the tropical climate (i.e., wet and dry Aw climate) is under the influence of the monsoon regime, dividing the year into two seasons: a dry season lasting from October to April,
and a rainy season lasting from May to October. The average annual rainfall in Lao PDR varies from 1,300 to 2,500 mm and can exceed 3,500 mm in central and southwestern Lao PDR (Nakhle et al., 2021).

The 0.6 km² Houay Pano headwater catchment (station S4) is located in northern Lao PDR, 10 km south of Luang Prabang city (Fig. 1a). This experimental site (Boithias et al., 2021b) is part of the Multiscale TROPIcal CatchmentS Critical Zone Observatory (M-TROPICS CZO; https://mtropics.obs-mip.fr/), a network of observatories under the French Research
Infrastructure OZCAR (Gaillardet et al., 2018). This catchment is representative of the montane agro-ecosystems of Southeast Asia. Altitude within the catchment is 435-716 m (Fig. 1b) and the slope gradient is 1-135 % (mean=52 %). Over the last decade, land-use change in the catchment mostly consisted of an increase of teak tree plantations at the expense of shifting cultivation (with slash-and-burn method). Overall, the areal percentage of annual crops decreased from 28% to virtually zero, while the areal percentage of teak tree plantations increased from 17% to 33% from 2011 to 2021.

### 2.2 Data collection

The dataset comprises *E. coli* concentrations ([*E. coli*]) with lower and upper limits of the confidence interval ([*E. coli*]$_{LL}$ and [*E. coli*]$_{UL}$, respectively), and physico-chemical measurements in stream water, recorded from May 25, 2011, to May 25, 2021. Physico-chemical measurements include temperature (T), electrical conductance (EC) at 25 °C, dissolved oxygen concentration ([DO]) and saturation (DO%), pH (pH), oxidation-reduction potential (ORP), turbidity (Turb), and total
suspended sediment concentration ([TSS]). The units of the eleven variables are given in Table 2.

For the spatial survey during both the dry and the rainy seasons, we chose sampling sites so as to ensure a broad geographical coverage of Lao PDR, and to represent a large range of geological, topographical, and land-use features. We also chose the sampling sites for being accessible from the road, in order achieve the sampling campaign within a relatively short time.



For the temporal monitoring, we initiated the sampling at stations S4, NK20, NK26, and A6, in May 2011 with a biweekly time-interval, changed to a 10-day time-interval in 2017. We initiated the monitoring at stations Nou_1, Nse_1, and MK_17 in July 2017 according to the 10-day time-interval. For MK_17, we used a boat to collect the water sample in the middle of the stream.

At the station S4, we collected water samples during flood events using an automatic sampler (Automatic Pumping Type Sediment Sampler, ICRISAT). The automatic sampler was triggered by a water level recorder to collect water after every 2

cm water level change during flood rising and every 4 cm water level change during flood recession. We measured EC, DO%, [DO], pH, and ORP once back to the laboratory, within 6 hours after automatic sampling, with a portable multi-probe system (YSI 556), and Turb with a portable turbidity meter (EUTECH Instruments TN-100).

At the other five stations, and at the station S4 during low flow, we manually sampled water with a bailer sampler and measured *in situ* T, EC, DO%, [DO], pH, and ORP with the multi-probe system and Turb with the turbidity meter. We stored water

samples in clean plastic bottles in an opaque icebox until laboratory analysis within 6 hours.

We measured [*E. coli*] in the laboratory with the standardized microplate method (ISO 9308-3). For each water sample, we incubated a water sub-sample at four dilution rates (i.e., 1:2, 1:20, 1:200, and 1:2000) in a 96-well microplate (MUG/EC, BIOKAR DIAGNOSTICS) for 48 h at 44 °C. Ringers' Lactate solution was used for the dilutions and one plate was used per sample. The number of positive wells for each microplate was noted and the Most Probable Number (MPN) per 100 mL was

determined using the Poisson distribution. Given the four dilution ratios, the detection limit of [*E. coli*] was 38 MPN 100 mL$^{-1}$ with lower confidence limit at 5.4 MPN 100 mL$^{-1}$ and upper confidence limit at 270 MPN 100 mL$^{-1}$.

We measured [TSS] in the laboratory after filtration on 0.2 μm porosity cellulose acetate filters (Sartorius) and evaporation at 105 °C for 48 h.

## 3 Results

The spatial survey (Ribolzi et al., 2021c) conducted in both the dry and the rainy seasons in 2016 resulted in 58 records (Fig. 2). The dataset shows contrasted values of [*E. coli*] depending on the season (Nakhle et al., 2021). Median [*E. coli*] are higher during the rainy season, similarly to [TSS], Turb, ORP, and T, while EC, DO%, and pH show smaller median values during the rainy season compared to the dry season.

The temporal monitoring at the outlet of the six catchments (Ribolzi et al., 2021a), initiated in May 2011 at stations NK20,

NK26, and A6 (Fig. 3), and in July 2017 at stations Nou_1, Nse_1, and MK_17 (Fig. 4), resulted in 1,131 records until May 2021. The dataset shows seasonal variations of [*E. coli*] and of the other variables (Boithias et al., 2016). In general, T and EC were increasing throughout the dry season and decreasing throughout the rainy season. The highest and lowest values of Turb, [TSS], and [*E. coli*], were measured during the rainy and the dry seasons, respectively. The temporal dynamics of DO%, [DO], pH, and ORP, are less clear. At the station NK26 the number of water samples with [*E. coli*] below the detection limit increased

after 2016 (Fig. 3). The data gap between March 12 and June 20, 2020, is due to COVID-19 lockdown and traffic restrictions.



The temporal monitoring at the station S4, outlet of the Houay Pano headwater catchment (Ribolzi et al., 2021b), initiated in May 2011, resulted in 413 records until May 2021 (Fig. 5). The dataset shows seasonal variations of [*E. coli*] and of the other variables (Boithias et al., 2016). Similar to the large catchments, T and EC were generally increasing throughout the dry season and decreasing throughout the rainy season. The highest and lowest values of Turb, [TSS], and [*E. coli*], were measured during

the rainy and the dry seasons, respectively. The temporal dynamics of DO%, [DO], pH, and ORP, are less clear. We monitored [*E. coli*] dynamics during 14 flood events (Boithias et al., 2021a). Values of [*E. coli*] measured during flood events with the automatic sampler are in general higher than those measured during low flow periods in grab samples.

## 4 Technical validation

We calibrated the multi-probe system each day before measurement, i.e. every day during the spatial survey (Ribolzi et al.,

2021c) and at a biweekly or 10-day frequency for the two other datasets (Ribolzi et al., 2021b, a). EC probe was calibrated with a 1,413 µS cm$^{-1}$ solution. [DO] probe was calibrated following the air-calibration chamber in air method. DO% was then automatically calibrated based on [DO] and the barometric pressure. pH probe was calibrated using a 3-point calibration (pH = 4.01, 7.01, and 9.18). ORP probe was calibrated with a 240 mV solution. For the turbidity meter, we verified each day before measurement that the turbidity measured using the 100 NTU calibration solution was correct. If a discrepancy was observed,

we calibrated the turbidity meter with 4 calibration solutions at 0.02, 20, 100, and 800 NTU. The accuracy of the suspended sediment mass on the 0.2 µm filters was ensured by the use of a 10$^{-4}$ g precision balance. We assessed the uncertainty of [*E. coli*] by using the MPN statistical method, which supplies the upper and the lower limits of the confidence interval.

We collected water samples as far as possible from the stream bank, to avoid any influence of the latter. However, we previously ensured that the values of T, EC, DO%, [DO], pH, ORP, Turb, [TSS], and [*E. coli*] measured at the sampling point

were representative of the variability along the stream transect.

Before laboratory analysis, water samples were stored in clean plastic bottles. These bottles were new empty bottles, supplied by a plastic bottle plant producing bottles for mineral water packaging. We regularly verified that the bottles were free from *E. coli*, and we triple rinsed the bottle with stream water before each water sampling.

Records were collated, curated, and cross-checked. If the information recorded in the databases was ambiguous or did not

match between records, we traced the sample back using the original paper records as far as possible. Samples which could not be verified in this way were excluded from publication in the current dataset.

## 5 Usage note

We published the three datasets as .CSV files so that they can be accessed and processed with any data processing software. The three datasets are open access, licensed under a Creative Commons Attribution 4.0 International License, with the

following additional requirements: all data users should (1) acknowledge the M-TROPICS CZO and its financing institutions





in their publications, (2) cite both the appropriate DOI of the data they used and the related publications, and (3) send to the corresponding authors of the present data description paper (LB and/or OR) a copy of any produced material based on the data. The corresponding authors, who are intimately familiar with the background of these datasets, are at the disposal of the authors wishing to reuse the datasets.

**6 Data availability**

The database includes a total of 1,602 records and is publicly available online as a collection of three files (Ribolzi et al., 2021b, a, c) hosted within the DataSuds platform (https://dataverse.ird.fr/). The three repositories are:

(1) *Escherichia coli* concentrations and physico-chemical measurements at the outlet of 29 catchments of the Mekong river basin, Lao PDR, during dry and rainy seasons (2016) (Ribolzi et al., 2021c): https://doi.org/10.23708/ZRSBM4

(2) *Escherichia coli* concentrations and physico-chemical measurements (2011-2021) at the outlet of six catchments of the Mekong river basin, northern Lao PDR (Ribolzi et al., 2021a): https://doi.org/10.23708/1YZQHH

(3) *Escherichia coli* concentrations and physico-chemical measurements (2011-2021) at the outlet of the Houay Pano catchment, northern Lao PDR (Ribolzi et al., 2021b): https://doi.org/10.23708/EWOYNK

We published the three datasets as .CSV files and a data dictionary along with the data files. The column headings of the data
dictionary files are listed in Table 2, together with the unit of each variable.

**7 Conclusions**

These three datasets together present a unique long-term spatiotemporal and multiscale surface water quality monitoring within the Mekong river basin. So far, the datasets have been used: (1) to describe the hydrological processes driving in-stream *E. coli* concentration during flood events (Boithias et al., 2021a; Ribolzi et al., 2016b), (2) to understand the role of land use in
bacterial dissemination on small and large catchment scales, e.g. *E. coli* (Causse et al., 2015; Rochelle-Newall et al., 2016; Nakhle et al., 2021; Ribolzi et al., 2011) and *Burkholderia pseudomallei* (Ribolzi et al., 2016a; Zimmermann et al., 2018; Liechti et al., 2021), (3) to relate stream water quality and diarrhea outbreaks (Boithias et al., 2016), and (4) to build catchment-scale numerical models focused on water quality (Kim et al., 2017, 2018; Abbas et al., 2021a, b).

The dataset may be further used (1) to assess the role of headwater catchments as *E. coli* source in large tropical river basins,
(2) to interpret new variables measured in the monitored catchments (e.g. contaminants other than *E. coli*), (3) to assess the impact of dams on downstream *E. coli* concentration, (4) to map the health risk posed by fecal pathogens, and (5) to assess the relative contributions of both climate and land-use change on changes in in-stream *E. coli* concentration.



**Author contribution**

O.R. designed the study. L.B., O.R., and A.P. coordinated the project. O.R., R.Z., S.R., P.O., and A.P. collected water samples
and performed field measurements for dataset https://doi.org/10.23708/ZRSBM4. L.B., O.R., C.T., B.S., K.L., N.Sil., P.S.,
K.X., J.C., and T.H.d.T collected water samples and performed field measurements for dataset
https://doi.org/10.23708/1YZQHH. L.B., O.R., E.R.N., C.T., B.S., K.L., N.Sil., P.S., K.X., T.P., O.E., and S.H. collected water
samples and performed field measurements for dataset https://doi.org/10.23708/EWOYNK. C.T. and A.P.B. performed
microbial laboratory analysis. L.B. and O.R. validated the data and curated the database. L.B., O.R., E.R.N., and P.N. analyzed
the data. O.S. and N.Sip. provided institutional support. L.B. wrote the original draft of the manuscript. All other authors
reviewed and edited the manuscript.

**Competing interests**

The authors declare that they have no conflict of interest.

**Acknowledgments**

These three datasets have been collected thanks to the long-term partnership with the Department of Agricultural Land
Management (DALaM), Lao PDR, which granted the permission for field access, and to the financial, scientific, technical,
and logistical support of the Multiscale TROPIcal CatchmentS Critical Zone Observatory (M-TROPICS CZO, previously
MSEC; https://mtropics.obs-mip.fr/).

The authors acknowledge the financial support of the Consultative Group on International Agricultural Research (CGIAR;
Humidtropics program; http://humidtropics.cgiar.org/), the GIS-Climat (Pastek program,
http://www.gisclimat.fr/projet/pastek.html), the French National Research Agency (TecItEasy project ANR-13-AGRO-0007;
https://anr.fr/), the Institut de Recherche pour le Développement (IRD), including through the International Joint Laboratory
on the impact of rapid Land-use change on Soil Ecosystem Services (LMI LUSES; ECOFILTER program; https://luses.ird.fr/)
and the regional pilot program Soils, Waters, Coastal Zones and Societies in Southern and Southeast Asia (SELTAR-RPP),
the International Center for Tropical Agriculture (CIAT) through the Payment for Environmental Services (PES) program in
Lao PDR, the French Centre National de la Recherche Scientifique (CNRS) through the national grants EC2CO-
Biohefect/Ecodyn//Dril/MicrobiEn (Belcrue, Belkong, and NARIBACT projects), the US Defence Threat Reduction Agency
Cooperative Biological Engagement Programme (contract HDTRA-16-C-0017), the Lao-Oxford-Mahosot Hospital-Wellcome
Trust Research Unit funded by the Wellcome Trust of Great Britain (Grant number 089275/H/09/Z), the Li Ka Shing
Foundation of the University of Oxford (grant SM40), and the Nam Theun 2 Power Company (NTPC).



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

## Figures



**Figure 1: Location of the 31 sampling stations within the Mekong river basin in Lao PDR. Tributary names and catchment areas are given in Table 1. Red dot represents sampling station S4 and green dots represent sampling stations NK20, NK26, A6, Nou_1, Nse_1, and MK_17.**


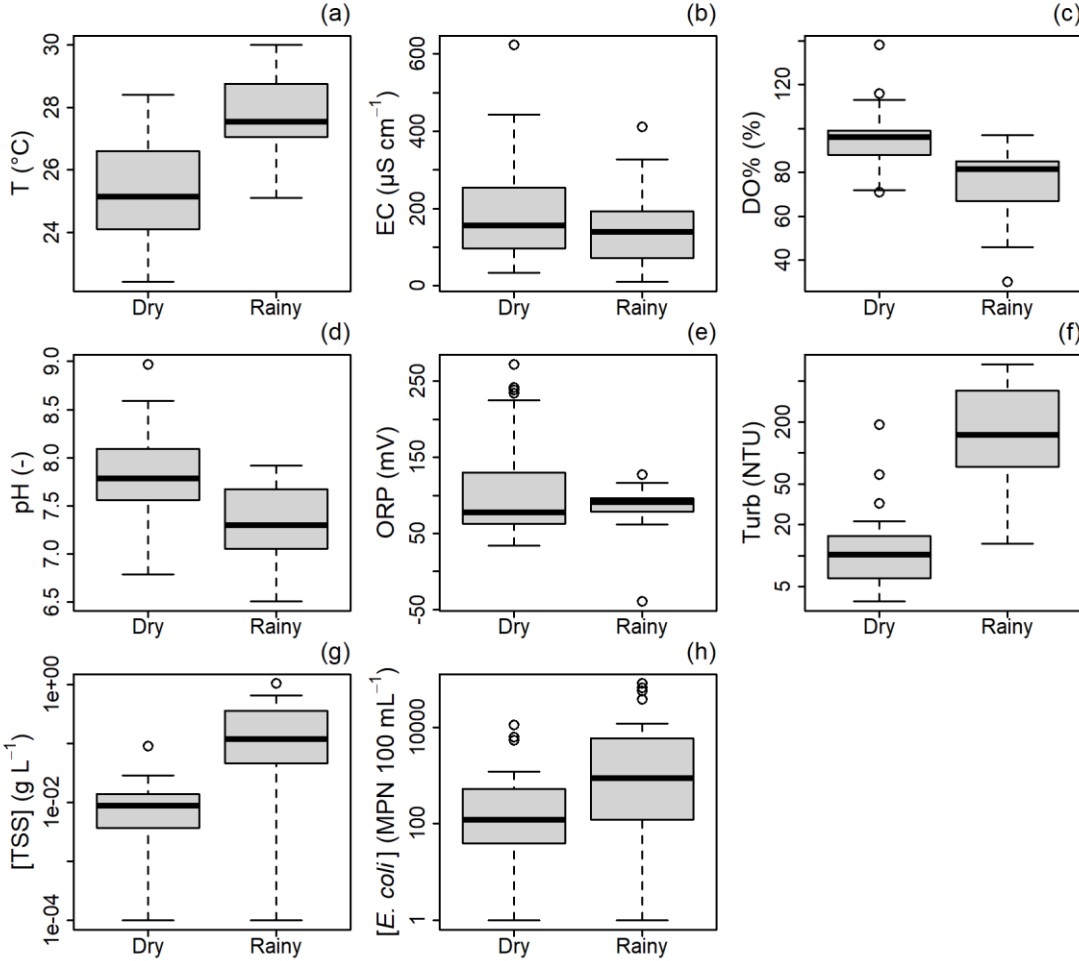

**Figure 2: Stream water quality during dry and rainy seasons in 2016 along the Mekong river (6 sampling stations) and in 19 of its tributaries (23 sampling stations) in Lao PDR. T: temperature (°C); EC: electrical conductance at 25°C (µS cm⁻¹); DO%: oxygen saturation (%); pH: pH (-); ORP: oxidation-reduction potential (mV); Turb: turbidity (NTU); [TSS]: total suspended sediment concentration (g L⁻¹); [E. coli]: Escherichia coli concentration (MPN 100 mL⁻¹). Turb, [TSS], and [E. coli] are shown with Y axis as log scale. We added 0.0001 and 1 to all [TSS] and [E. coli] values, respectively, to present 0-values in a log scale, by convention.**


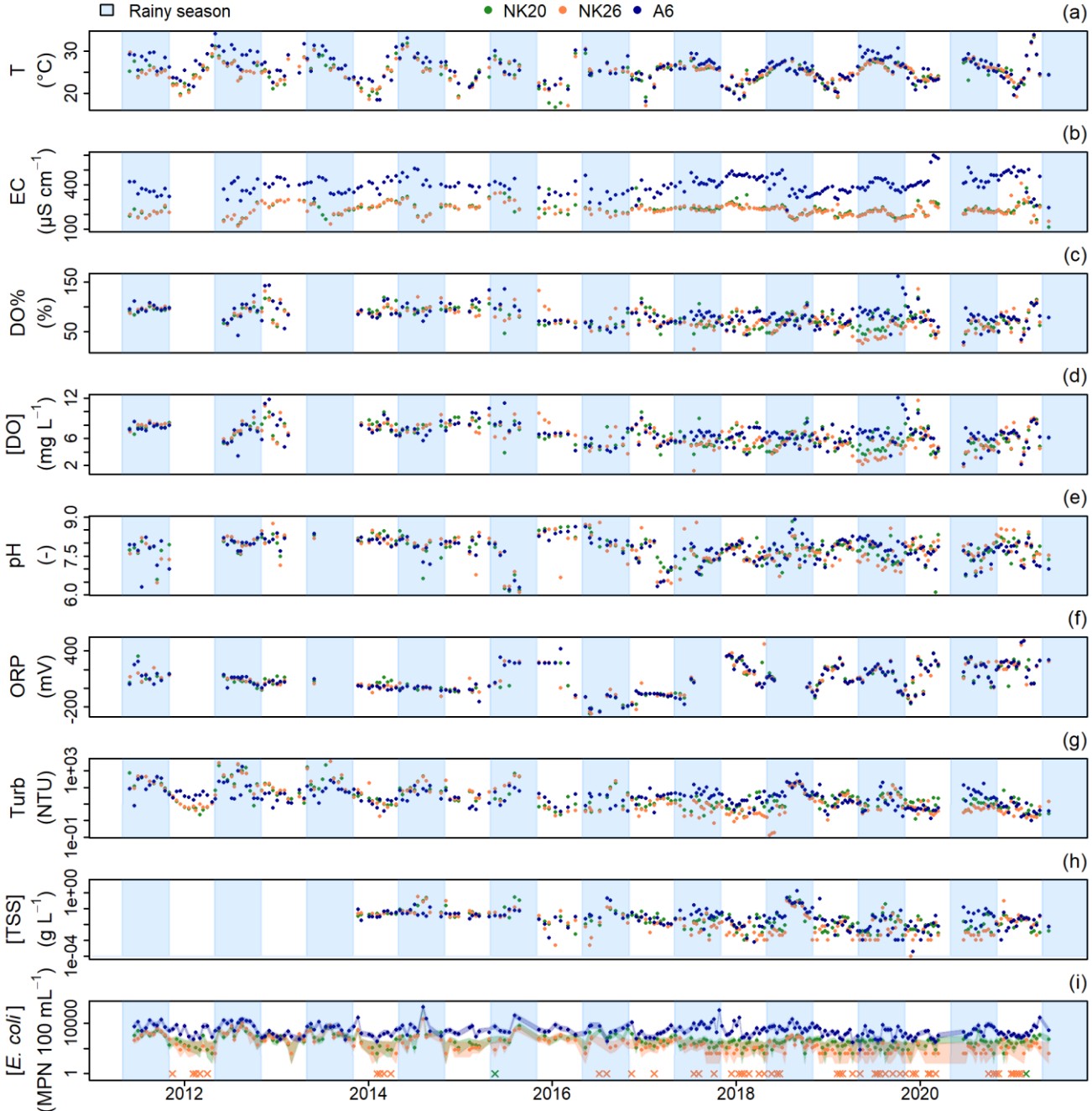

**Figure 3: Stream water quality from 2011 to 2021 along the Nam Khan river (sampling stations NK20 and NK26) and its tributary Houay Khan (sampling station A6), northern Lao PDR. T: temperature (°C); EC: electrical conductance at 25°C (µS cm⁻¹); DO%: oxygen saturation (%); [DO]: oxygen concentration (mg L⁻¹); pH: pH (-); ORP: oxidation-reduction potential (mV); Turb: turbidity (NTU); [TSS]: total suspended sediment concentration (g L⁻¹); [E. coli]: *Escherichia coli* concentration (MPN 100 mL⁻¹) with lower and upper limits of the confidence interval given by Poisson distribution using the standardized microplate method. Turb, [TSS], and [E. coli] are shown with Y-axis as log scale. We added 0.0001 and 1 to all [TSS] and [E. coli] values, respectively, to present 0-**





values in a log scale, by convention. Crosses for [*E. coli*] represent [*E. coli*] below the detection limit. Blue polygons represent the rainy season from May to October.

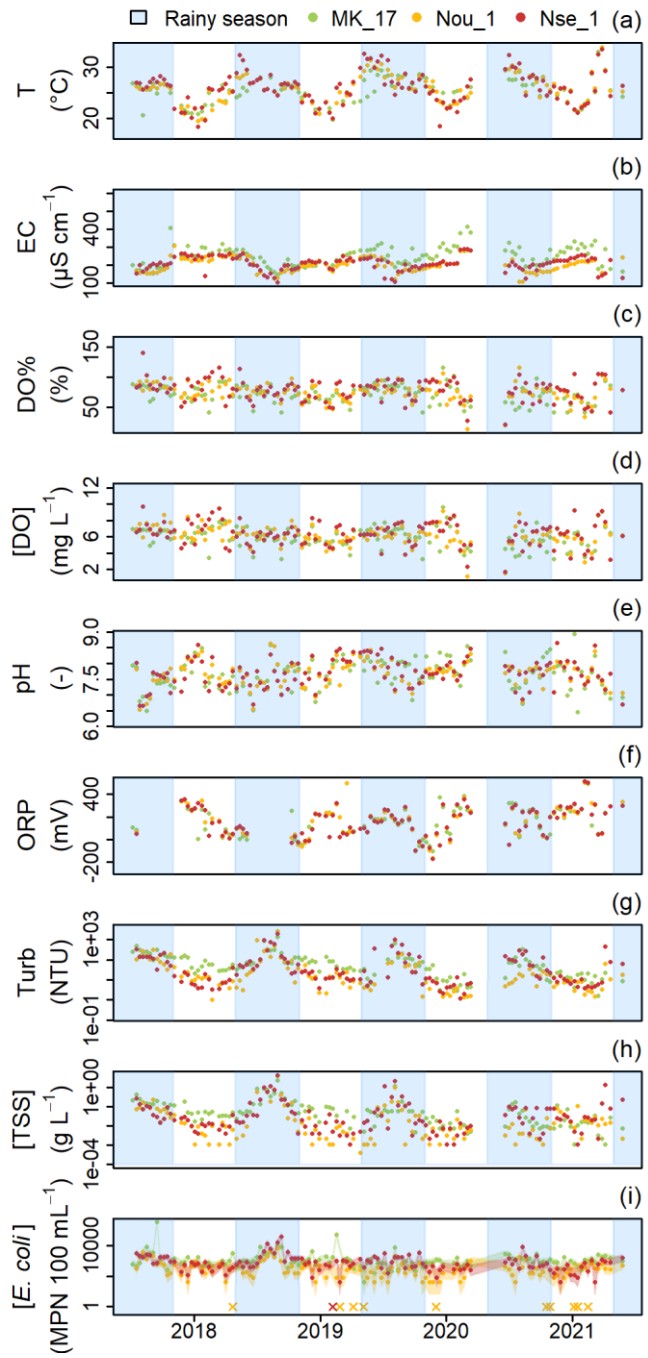


**Figure 4: Stream water quality from 2017 to 2021 in the Mekong at Luang Prabang (sampling station MK_17) and its tributaries Nam Ou and Nam Seuang (sampling stations Nou_1 and Nse_1, respectively), northern Lao PDR. T: temperature (°C); EC: electrical conductance at 25°C (µS cm⁻¹); DO%: oxygen saturation (%); [DO]: oxygen concentration (mg L⁻¹); pH: pH (-); ORP: oxidation-reduction potential (mV); Turb: turbidity (NTU); [TSS]: total suspended sediment concentration (g L⁻¹); [E. coli]: *Escherichia coli*** 

**concentration (MPN 100 mL⁻¹) with lower and upper limits of the confidence interval given by Poisson distribution using the standardized microplate method. Turb, [TSS], and [E. coli] are shown with Y axis as log scale. We added 0.0001 and 1 to all [TSS]**



**and [*E. coli*] values, respectively, to present 0-values in a log scale, by convention. Crosses for [*E. coli*] represent [*E. coli*] below the detection limit. Blue polygons represent the rainy season from May to October.**



**Figure 5: Stream water quality from 2011 to 2021 at the outlet of the Houay Pano catchment (sampling station S4), northern Lao PDR. T: temperature (°C); EC: electrical conductance at 25°C (µS cm⁻¹); DO%: oxygen saturation (%); [DO]: oxygen concentration (mg L⁻¹); pH: pH (-); ORP: oxidation-reduction potential (mV); Turb: turbidity (NTU); [TSS]: total suspended sediment concentration (g L⁻¹); [E. coli]:** *Escherichia coli* **concentration (MPN 100 mL⁻¹) with lower and upper limits of the confidence interval given by Poisson distribution using the standardized microplate method. Turb, [TSS], and [E. coli] are shown with Y axis as log scale. We added 0.0001 and 1 to all [TSS] and [E. coli] values, respectively, to present 0-values in a log scale, by convention. Crosses for [E. coli] represent [E. coli] below the detection limit. Blue polygons represent the rainy season from May to October.**





**Tables**

**Table 1: Description of the 31 sampling stations within the Mekong river basin in Lao PDR: sampling station name, river name, geographical coordinates of sampling station (i.e., latitude and longitude in degrees, WGS 1984), sampling period, and catchment drainage area in km².**

| Sampling station | River | Latitude (°) | Longitude (°) | Sampling period | Catchment area (km²) |
|---|---|---|---|---|---|
| S4 | Houay Pano | 19.85262 | 102.16912 | 2016, 2011-2021 | 0.6 |
| NK26 | Nam Khan | 19.75536 | 102.21755 | 2011-2021 | 6,885 |
| NK20 | Nam Khan | 19.78602 | 102.18336 | 2016, 2011-2021 | 7,236 |
| A6 | Houay Khan | 19.76010 | 102.18112 | 2016, 2011-2021 | 239 |
| Nou_1 | Nam Ou | 20.08642 | 102.26406 | 2016, 2017-2021 | 25,946 |
| Nse_1 | Nam Seuang | 19.97931 | 102.24728 | 2016, 2017-2021 | 6,577 |
| MK_17 | Mekong | 19.89267 | 102.13074 | 2017-2021 | 272,155 |
| Npa_1 | Nam Pa | 19.96049 | 102.28289 | 2016 | 758 |
| Nmi_1 | Nam Mi | 17.91917 | 101.68856 | 2016 | 1,021 |
| Nsa_1 | Nam Sang | 18.22284 | 102.14222 | 2016 | 1,210 |
| Ntho_1 | Nam Thôn | 18.09152 | 102.28159 | 2016 | 582 |
| Nlik_1 | Nam Lik | 18.63280 | 102.28104 | 2016 | 3,022 |
| Nng_3 | Nam Ngum | 18.52502 | 102.52631 | 2016 | 8,366 |
| Nng_4 | Nam Ngum | 18.35581 | 102.57204 | 2016 | 14,318 |
| Nng_2 | Nam Ngum | 18.20269 | 102.58588 | 2016 | 14,985 |
| Nng_1 | Nam Ngum | 18.17879 | 103.05593 | 2016 | 16,841 |
| Nma_1 | Nam Mang | 18.37019 | 103.19846 | 2016 | 1,793 |
| Ngn_1 | Nam Gniep | 18.41756 | 103.60217 | 2016 | 4,564 |
| Nxa_1 | Nam Xan | 18.39523 | 103.65408 | 2016 | 2,223 |
| Nka_1 | Nam Kadin | 18.32517 | 103.99924 | 2016 | 14,820 |
| Nhi_1 | Nam Hin Boun | 17.72699 | 104.56798 | 2016 | 2,152 |
| Xbi_3 | Xe Bang Fai | 17.07782 | 104.98496 | 2016 | 9,433 |
| Xbg_1 | Xe Bang Hieng | 16.09804 | 105.37625 | 2016 | 19,817 |
| Xbn_1 | Xe Bang Nouan | 16.00290 | 105.47937 | 2016 | 1,351 |
| SR_1 | Nam Sedon | 15.12390 | 105.80748 | 2016 | 7,225 |
| MK_1 | Mekong | 19.95601 | 102.24113 | 2016 | 263,880 |
| MK_7 | Mekong | 17.89870 | 101.62397 | 2016 | 295,246 |
| MK_2 | Mekong | 17.97276 | 102.50410 | 2016 | 301,826 |
| MK_3 | Mekong | 17.39714 | 104.79999 | 2016 | 373,368 |
| MK_4 | Mekong | 16.00503 | 105.42449 | 2016 | 417,094 |
| MK_5 | Mekong | 15.10721 | 105.79878 | 2016 | 549,055 |



**Table 2: Description of the 18 column headings of the database files along with variables units.**

| Variable | Column heading | Description | Unit |
|---|---|---|---|
| | Outlet | Name of the sampling station | - |
| | LAT | Latitude, geographical coordinates of the sampling station | ° |
| | LONG | Longitude, geographical coordinates of the sampling station | ° |
| | River | Name of the river | - |
| | Date | Day of sampling | - |
| | Time | Time of sampling | - |
| | Date_Time | Day and time of sampling | - |
| T | T | Stream water temperature | °C |
| EC | EC | Stream water electrical conductance à 25°C | $\mu S\ cm^{-1}$ |
| DO% | DOpercent | Stream water dissolved oxygen saturation | % |
| [DO] | DO | Stream water dissolved oxygen concentration | $mg\ L^{-1}$ |
| pH | pH | Stream water pH | - |
| ORP | ORP | Stream water oxidation-reduction potential | mV |
| Turb | Turbidity | Stream water turbidity | NTU |
| [TSS] | TSS | Stream water total suspended sediment concentration | $g\ L^{-1}$ |
| $[E.\ coli]_{LL}$ | E-coli_4dilutions_95%-CI-LL | Lower limit of the confidence interval of the *Escherichia coli* concentration in water | $MPN\ 100\ mL^{-1}$ |
| $[E.\ coli]$ | E-coli_4dilutions | Stream water *Escherichia coli* concentration | $MPN\ 100\ mL^{-1}$ |
| $[E.\ coli]_{UL}$ | E-coli_4dilutions_95%-CI-UL | Upper limit of the confidence interval of the *Escherichia coli* concentration in water | $MPN\ 100\ mL^{-1}$ |
