# Peer review of "Escherichia coli* concentration, multiscale monitoring over the decade 2011-2021 in the Mekong basin, Lao PDR"

_Earth System Science Data, 2021_

## Author Response (AR1)

Dear Editor,

Please find attached the revised version of our manuscript '***Escherichia coli* concentration, multiscale monitoring over the decade 2011-2021 in the Mekong basin, Lao PDR**'.

We would like to thank you for your interest in our work, and we hope that this new version will be considered acceptable for review in the *Earth System Science Data* journal.

Yours sincerely,

Laurie BOITHIAS & Olivier RIBOLZI

E-mail: laurie.boithias@get.omp.eu & olivier.ribolzi@ird.fr

\*\*\*

**Please note that line numbers refer to the track-change version of the manuscript.**

**Topical Editor comments to the author:**

**[TEC1]** In usage note, authors specify CC-BY license but then add conditions that seem more like CC-BY-SA. Data journals do not allow -SA; such requirements defeat our purpose. We will see how reviewers respond. Authors should prepare to 'relax' these supplemental requirements (e.g recommend vs. require). Uncertain about ESSD reviewers will react to this product. We will find out...

Thank you for your comment. It seems that the reviewers did not comment on this point. However, we rephrased the two sentences at L172-173 as such:

"The three datasets are open access, licensed under a Creative Commons Attribution 4.0 International License. For the proper functioning and the sustainability of the CZO, M-TROPICS asks data users to (1) …"

**Referees' comments to the author:**

**[R1C1]** The dataset is a unique collection of multiscale water quality measurements for a tropical watershed. It does not have analogs. By making it available, the authors have made an outstanding contribution to hydrology.

We thank the referee for her/his kind comment.

**[R1C2]** Some clarifications need to be made. The authors state that they "previously ensured that the values of T, EC, DO%, [DO], pH, ORP, Turb, [TSS], and [E. coli] measured at the sampling point was representative of the variability along the stream transect." The methodology has to be described or references need to be given.

We checked the variability of the variables' values along the river cross-sectional profile of station NK20, which is a referenced station of the Lao PDR national hydrological monitoring network. We measured T, EC, DO%, [DO], pH, ORP, Turb, and [*E. coli*] at a distance of 0.5, 15, 30, 45 and 68 m

away from the left bank, and at a depth of 0.5, 1, and 2 m from the water surface. Sampling was done in June in order to have a transient hydrological regime between dry season (low flow) and rainy season (high flow). A total of 10 samples were collected across the section. We found that the values observed at the long-term monitoring sampling point (typically between 5 and 10 m from the left bank) was within the range of variation (min, max) given by the samples across the section.

We made this dataset available within the DataSuds platform (https://dataverse.ird.fr/) with following DOI: https://doi.org/10.23708/RNY0LD.

We changed the sentence at L156-162 to:

"We collected water samples as far as possible from the stream bank, to avoid any influence of the latter. Similarly, we chose sampling stations so as to not be affected by an upstream confluence. Physico-chemistry and suspended sediment concentration can be heterogeneous along a stream cross-section (Santini et al., 2019), but for reasons of logistical capacity of sampling, the measurements could only be made from a single sample at the different stations. However, to ensure that measurements at sampling point were within the range of variation of the values measured along the stream cross-section, we performed a 10-point survey across the river section at station NK20, which is a referenced station of the Lao PDR national hydrological monitoring network (Ribolzi et al., 2022)."

We updated the references' list accordingly.

We also added the following sentence at L110-113:

"Bailer-sampled water was typically collected 5-10 m from the stream bank, except for station S4 where water was collected in the middle of the stream, about 0.5 m from the stream bank, and for stations MK_17, MK_3, and MK_5, where we used a boat to collect the water sample in the middle of the stream."

[R1C3] The authors state that "[DO] probe was calibrated following the air-calibration chamber in air method" The method has to be described or referenced.

We added the reference to the sentence which is now on L148:

"[DO] probe was calibrated following the air-calibration chamber in air method (USGS, 2006)."

We updated the references' list accordingly.

[R1C4] The duration of the storage and its effect on the water quality has to be described.

The maximal duration of storage before laboratory analysis was 6 hours (L106 and L114). We stored water samples in an opaque icebox until laboratory analysis. The effect of storage is limited: there is no significant variation of [*E. coli*] within 6 hours (Nakhle et al., 2021). We added the following sentence on L107-108:

"Maximal storage duration of 6 hours ensured that [*E. coli*] variation was not significant before laboratory analysis (Nakhle et al., 2021a)."

We updated the references' list accordingly.

[R2C1] This unique and detailed dataset on water quality measurements in the Mekong River is a valuable contribution to the field of health-related water microbiology and hydrology and the

authors make it freely available for future use by others. It regroups data collected over a decade at various spatial and temporal scales, and part of the data presented here have been focused on in recent publications of the group.

We thank the referee for her/his kind comment.

[R2C2] Line 98: please add "to" before "achieve".

We corrected it on L98.

[R2C3] Line 111: Please specify whether you performed blanks for E. coli enumeration to ensure absence of contamination. Considering the LOD determined here, it could be useful to indicate in the text the percentage of samples below LOD.

We used sterile Ringers' Lactate solution for the dilutions with the microplate *E. coli* enumeration method. We performed blanks on Ringers' Lactate solution several times over the decade, although we did not test it routinely. We added the following sentence on L118:

"We tested the sterility of Ringers' Lactate solution and found no *E. coli*."

As for the percentage of samples below LOD, we added the following sentence on L121-122:

"Among the three datasets, the number of water samples with [*E. coli*] below the limit of detection was 5.9 %."

[R2C4] Line 117: what volumes were typically filtered for TSS measurements? Considering the very small porosity of the filters (0,2 um), I would expect (very) small volumes to be filtered, which could in turn introduce a source of uncertainty. Usually, TSS is determined using glas-fiber filters of 1.2 - 1.5 um nominal porosity. Please explain how you proceeded here, especially with turbidities exceeding 200 NTU. Please cite an appropriate SOP reference.

The volume of water filtered for [TSS] measurement was 100 mL. We used 0.2 µm filters to ensure that filtration traps the clay fraction (<2 µm) of the water samples, knowing that clay fraction is up to 72.9 % in adjacent soils (Chaplot and Poesen, 2012). We rephrased the sentence as such on L123-125:

"We measured [TSS] in the laboratory after the filtration of 100 mL of sample water on 0.2 µm porosity cellulose acetate filters (Sartorius) and evaporation at 105 °C for 48 h. We used 0.2 µm filters to ensure that filtration trapped the clay fraction of the water samples, knowing that clay fraction is up to 72.9 % in adjacent soils (Chaplot and Poesen, 2012)."

We updated the references' list accordingly.

---

## Author Response (AR2)

Dear Editor,

Please find attached the revised version of our manuscript '***Escherichia coli* concentration, multiscale monitoring over the decade 2011-2021 in the Mekong basin, Lao PDR**'.

We would like to thank you for accepting our manuscript for publication in the *Earth System Science Data* journal.

Yours sincerely,

Laurie BOITHIAS & Olivier RIBOLZI

E-mail: laurie.boithias@get.omp.eu & olivier.ribolzi@ird.fr
* * *
1. Please note that the last names of the authors should not be all capitalized.

We rewrote the last names of the authors with first letter capitalized only.

2. Regarding your figure #3, #4: please ensure that the colour schemes used in your maps and charts allow readers with colour vision deficiencies to correctly interpret your findings. Please check your figures using the Coblis – Color Blindness Simulator (https://www.color-blindness.com/coblis-color-blindness-simulator/) and revise the colour schemes accordingly.

We checked both Fig. 3 and 4 in COBLIS. We found that the colors in Fig. 4 had to be improved and we changed the yellowish color to a lighter one, which makes the Fig. 4 now suitable for readers with color vision deficiencies according to COBLIS.

Please note that by re-producing Fig. 4 we changed the number format for *E. coli* plot from standard to scientific format. We therefore also updated the Fig. 2, 3, and 5.